# Direct Pellet Three-Dimensional Printing of Polybutylene Adipate-co-Terephthalate for a Greener Future

**DOI:** 10.3390/polym16020267

**Published:** 2024-01-18

**Authors:** Armin Karimi, Davood Rahmatabadi, Mostafa Baghani

**Affiliations:** 1School of Mechanical Engineering, College of Engineering, University of Tehran, Tehran P.O. Box 11155-9567, Irand.rahmatabadi@ut.ac.ir (D.R.); 2Department of Aerospace Engineering, Sharif University of Technology, Tehran P.O. Box 11155-9567, Iran

**Keywords:** PBAT, 3D printing, nozzle temperature, material extrusion, biodegradable plastics, mechanical properties

## Abstract

The widespread use of conventional plastics in various industries has resulted in increased oil consumption and environmental pollution. To address these issues, a combination of plastic recycling and the use of biodegradable plastics is essential. Among biodegradable polymers, poly butylene adipate-co-terephthalate (PBAT) has attracted significant attention due to its favorable mechanical properties and biodegradability. In this study, we investigated the potential of using PBAT for direct pellet printing, eliminating the need for filament conversion. To determine the optimal printing temperature, three sets of tensile specimens were 3D-printed at varying nozzle temperatures, and their mechanical properties and microstructure were analyzed. Additionally, dynamic mechanical thermal analysis (DMTA) was conducted to evaluate the thermal behavior of the printed PBAT. Furthermore, we designed and printed two structures with different infill percentages (40% and 60%) to assess their compressive strength and energy absorption properties. DMTA revealed that PBAT’s glass–rubber transition temperature is approximately −25 °C. Our findings demonstrate that increasing the nozzle temperature enhances the mechanical properties of PBAT. Notably, the highest nozzle temperature of 200 °C yielded remarkable results, with an elongation of 1379% and a tensile strength of 7.5 MPa. Moreover, specimens with a 60% infill density exhibited superior compressive strength (1338 KPa) and energy absorption compared with those with 40% infill density (1306 KPa). The SEM images showed that with an increase in the nozzle temperature, the quality of the print was greatly improved, and it was difficult to find microholes or even a layered structure for the sample printed at 200 °C.

## 1. Introduction

Owing to the advancement of the economy and living standards in the 21st century, conventional plastics have been extensively used in various fields due to their favorable properties and low costs [1]. However, these plastics, derived from petroleum, cannot be degraded, leading to increased oil consumption and environmental pollution. To address these issues, the recycling of conventional plastics and the use of biodegradable plastics should be pursued together [2]. Biodegradable plastics can decompose naturally with the help of microorganisms, making them a beneficial supplement to plastic recycling [3]. In order to solve environmental problems, there is a growing interest in designing new biodegradable polymers, particularly polyesters. Aliphatic–aromatic co-polyesters, including poly butylene adipate-co-terephthalate (PBAT), have exhibited considerable potential due to their beneficial mechanical properties and biodegradability [4]. PBAT is a widely recognized biodegradable plastic material. Notably, PBAT boasts excellent flexibility, high elongation at break, remarkable hydrophilic characteristics, and favorable processing properties [5]. Its ability to degrade under both composting and soil conditions positions PBAT as a viable solution to combat plastic pollution [6]. Consequently, PBAT has predominantly found application in the production of environmentally friendly plastic films, contributing to the pursuit of sustainable alternatives to conventional plastics [7].

The advancement of 3D printing technology has generated significant interest among researchers, particularly in the realm of polymer production [8,9]. This has consequently led to the development of various polymer-manufacturing techniques within the field of 3D printing, including, but not limited to, selective laser sintering (SLS), stereolithography (SLA), digital light processing (DLP), polyjet/multijet modeling (PJM/MJM), and fused deposition modeling (FDM) [10,11,12,13]. It provides great design flexibility and can produce complex geometries that are difficult to achieve with traditional manufacturing techniques [14,15]. Despite advancements in 3D printing technology, the majority of materials utilized for the FDM process, except for PLA, are non-biodegradable polymers, like acrylonitrile butadiene styrene (ABS), polyamides (PA), and polypropylene (PP) [16]. PLA and polycaprolactone (PCL) are biodegradable polymers commonly used in FDM 3D printing [17]. The development of biodegradable materials specifically designed for 3D printing is imperative in order to facilitate the transition toward sustainable and environmentally friendly 3D printing practices. One such material is PBAT. This polymer possesses several desirable properties that render it highly suitable for 3D printing applications. Notably, it exhibits excellent thermal stability and mechanical characteristics. Moreover, it can be blended with conventional polymers, like poly(lactic acid) (PLA) [18], poly(3-hydroxybutyrate-co-3-hydroxyvalerate) (PHBV) [19], poly(propylene carbonate) (PPC) [20], polyvinyl chloride (PVC) [21], and PCL [22], effectively improving their biodegradability. PBAT offers complete biodegradability and composability, enabling the 3D printed components to naturally degrade within municipal and industrial composting facilities, as well as natural environments. This distinctive capability enables the exploration of various potential uses for PBAT-based 3D printed objects, including biomedical implants, food packaging, and disposable consumer products. The integration of PBAT into 3D printing can, therefore, promote the adoption of sustainable practices and reduce the ecological footprint associated with this rapidly growing industry.

According to a review of the literature, PBAT has been widely used in the form of a blend with other polymers to be used as a 3D printing material. Andrzejewski et al. [23] explored the modification of PLA for 3D printing (FDM printing technique) purposes through blending it with PBAT. By incorporating PBAT in different percentages, it was found that the impact strength of PLA significantly improved. The addition of a multifunctional chain extender (ESA) also improved the rheological properties of the PLA/PBAT material, making it easier to produce filaments and maintain a consistent printed material path. The thermomechanical analysis showed no significant decrease in the thermal resistance of the materials, while there was evidence of PLA structure nucleation by PBAT. Comparing printed and injected PLA/PBAT blends indicated that ESA-modified samples had similar or better mechanical properties. Zhang et al. [24] conducted a study focused on the development and use of biodegradable polymers, specifically polyglycolic acid (PGA) and PBAT, as feedstocks for 3D printing. The researchers found that blends of PGA and PBAT exhibited desired mechanical properties, such as stiffness and toughness, when formulated in specific ratios. These blends were then successfully 3D printed using a fused deposition modeling (FDM) technique. The resulting printed structures showed comparable properties to injection-molded samples, demonstrating the feasibility and potential of using biodegradable polymers for 3D printing. Moreover, they were able to manufacture industrial parts with excellent dimensional stability and quality using this biodegradable material.

To the best of our knowledge, pure PBAT has not yet been explored as a standalone material for 3D printing applications, primarily due to its soft and elastic nature [25]. Soft materials inherently pose several challenges during the 3D printing process [26]. For instance, when employing the FDM technique—widely adopted in the field of 3D printing—issues like filament buckling and filament misalignment commonly arise when dealing with soft materials, rendering conventional FDM impractical for printing soft materials, like PBAT. These issues arise from the lack of any lateral support for the filament between the filament feed and the hotend entrance in the extruder [27]. Blending PBAT with other polymers has been suggested as a solution to the challenges faced by PBAT 3D printing. However, studies have revealed that pure PBAT exhibits superior biodegradability compared with its blended counterparts [5]. Another potential solution proposed for printing soft materials involves eliminating filament and directly utilizing PBAT pellets for printing purposes [28]. This approach offers several benefits: firstly, the resulting object is entirely composed of PBAT, a bio-based and eco-friendly material that readily decomposes in the natural environment, thus minimizing its impact on its surroundings. Secondly, by eliminating the need for an intermediary filament, this technique streamlines the printing process, enhances material compatibility, and reduces production costs, making it a promising and sustainable alternative for fabricating soft objects.

This study used direct pellet printing technology for the 3D printing of PBAT structures. Our aim was to examine the feasibility of printing PBAT parts without the prerequisite of converting pellets into filaments. To accomplish this, we printed three sets of tensile specimens with varying nozzle temperatures to determine the optimal temperature by investigating their mechanical properties and microstructure. Additionally, we used dynamic mechanical thermal analysis (DMTA) for investigating the thermal behavior of the printed PBAT. Furthermore, we designed and printed two energy-absorption structures with optimized parameters, comprising two different infill percentages to study the compressive and energy-absorption behavior of the material. This direct pellet printing approach could potentially simplify PBAT 3D printing and reduce material waste compared with conventional filament extrusion methods.

## 2. Experimental Procedures

### 2.1. Material

PBAT pellets of KD 1024 grade were purchased from the Zhuhai Wango Chemical Co. (Zhuhai, China), boasting a specific gravity of 1.21–1.23 g/cm^3^ and a flow index of 4 g/10 min at 190 °C. These pellets served as the primary raw material for direct pellet printing in this study.

### 2.2. Printing Process

The material extrusion process in additive manufacturing is classified in 3 methods based on the extrusion method. These 3 methods are shown in Figure 1. In our research, we employed a printer specifically designed for the printing of PBAT pellets. To address the problem of filament buckling commonly encountered in standard fused filament fabrication printers (Figure 1a), we customized this printer with alterations, making it distinct. By substituting the original filament-feeding mechanism with a pneumatically operated pellet hopper and feeding system (Figure 1b), we ensured accurate and controlled feeding of polymer pellets. This modification facilitated the seamless feeding of pellets into the extruder, where they were melted and expelled through a heated nozzle.

### 2.3. Dynamic Mechanical Thermal Analysis

DMTA was conducted in this study to assess the thermomechanical properties and identify the distinct thermal zones of PBAT. The DMTA testing was performed using a dynamic mechanical thermal analyzer (Mettler Toledo, Greifensee, Switzerland). The temperature ranges for the analysis spanned from −100 °C to 120 °C, with a heating rate of 5 °C/min and a constant frequency of 1 Hz. A cantilever beam with dimensions of 40 mm × 10 mm × 1 mm was prepared in accordance with the tensile mode requirements outlined in the ASTM D4065-01 standard. This specific geometry was employed to ensure accurate and consistent measurement of the thermomechanical properties of the printed PBAT.

### 2.4. Mechanical Properties

Mechanical properties evaluation of the 3D printed parts was carried out. Tensile specimens were printed based on the dimensions outlined in ASTM D412-C with the printing parameters shown in Table 1. Tensile specimens were printed with 3 different nozzle temperatures to study the effect of this parameter on mechanical properties. The specimens were tested under uniaxial tension using a SANTAM STM50 mechanical tester (Tehran, Iran) at a strain rate of 100 mm/min until failure. The strain and stress were recorded for at least 3 specimens under each printing condition and the averages of the values were reported. The aim of the tensile characterization was to assess the stretchability and strength of the printed parts. In addition, the desirable mechanical properties could prove the quality of printed parts. Figure 2 depicts the printed samples and their dimensions.

### 2.5. Microstructure

The microstructure and layer adhesion of the 3D printed PBAT parts were investigated using scanning electron microscopy (SEM). A Vegall model SEM instrument was employed for this purpose. To prepare the samples for imaging, freeze fracturing was performed at temperatures below −50 °C. Prior to imaging, the freeze-fractured samples were gold-coated to enhance their conductivity.

### 2.6. Compressive and Energy-Absorption Behavior

In order to investigate the energy absorption capabilities of the printed structures, we designed and manufactured 2 cubic specimens. These specimens were specifically constructed with optimized printing parameters and varying infill percentages (60% in Figure 3a and 40% in Figure 3b) to assess their respective performances. The specimens were tested under uniaxial compression using a SANTAM STM50 mechanical tester (Tehran, Iran) at a strain rate of 20 mm/min.

To visualize the actual printed structures before, during, and after the subsequent energy absorption tests, refer to Figure 4. This figure illustrates the details of the testing process and setups.

### 2.7. Ability to Maintain Shape

To ensure the precision of the 3D printed cubic samples, we assessed their dimensions by comparing them with a CAD model. To address the issue of material shrinkage during the cooling process, the expansion parameter (EP) comes into play. This parameter allows for the size adjustment of the printed parts in the x and y directions. If the printed part’s dimensions are smaller than expected due to shrinkage, a positive EP value increases the outside dimensions. Conversely, a negative value decreases the outside dimensions if the printed model exceeds the anticipated size. The EP parameter serves as a means to fine-tune the printed parts and maintain the desired dimensional accuracy [30].

## 3. Results and Discussion

### 3.1. Dynamic Mechanical Thermal Analysis

The DMTA responses of the printed PBAT were analyzed in terms of storage modulus and tan δ, and the results are shown in Figure 5. In this figure, the material exhibited a glassy state between −100 °C and about −50 °C, with the highest and most stable storage modulus value of 1700 MPa. The applied oscillatory deformation exceeded the relaxation time of the side-group polymer segments, resulting in the material exhibiting a rigid behavior [31]. Within the temperature range of −50 °C to 0 °C, the storage modulus rapidly decreased to 103 MPa, indicating the onset of the glass transition region. This decrease was likely due to the weakening of molecular bonds in the polymer chains caused by the elevated thermal energy during the transition phase. Beyond this point, the material remained in a rubbery state. The applied oscillatory deformation was far slower than the cooperative segmental movements; thus the internal reorganizations elastically absorbed the solicitation. Thus, E’ showed a constant value that may have been related to the molecular weight between entanglements or crosslinks [32]. This region was called the rubbery plateau.

As depicted in Figure 5, PBAT is in its rubbery phase at room temperature, with a storage modulus of 61 MPa. This characteristic leads to an elastomeric and soft behavior of the material at room temperature, which poses challenges when attempting to print PBAT using the conventional FFF process.

Tan δ serves as a measure of the energy dissipation potential in a viscoelastic material, indicating the ratio of the viscous to elastic response [33]. When a load is applied to a polymer, some of the load is dissipated through mechanisms such as polymer chain segmental motion, while the remaining energy is stored in the material and eventually releases upon load removal, akin to the response of a spring. The height and area under the tan δ curve provide insights into the total amount of energy that can be absorbed by the material, with a larger area indicating higher molecular mobility and improved damping properties.

For materials aimed at better impact absorption, increasing the area under the tan δ curve is desirable. Figure 5 illustrates a peak in the tan δ diagram, indicating the glass transition temperature of PBAT, which is estimated to be around −25 °C. Previous studies have also reported similar results [34]. Moreover, the area under the tan δ curve is greater after the glass transition, indicating improved energy absorption characteristics.

### 3.2. Mechanical Properties

Figure 6 presents the stress–strain curves of the printed dogbone samples with varying nozzle temperatures. The results indicate that, as the nozzle temperature increased, the mechanical properties of the printed materials also improved. For instance, the sample printed at a nozzle temperature of 160 °C exhibited approximately 446% elongation and 6.5 MPa tensile strength. These values increased to approximately 586% elongation and 7.5 MPa tensile strength when the nozzle temperature was raised to 180 °C. Interestingly, the sample printed at the highest nozzle temperature of 200 °C demonstrated the best mechanical properties, displaying a remarkable elongation of 1379% and a tensile strength of 7.5 MPa. It should be noted that this elongation at break surpassed results reported in previous studies on PBAT, while the tensile strength remained comparatively lower. In these studies [35,36,37], PBAT elongation was reported to be about 700% to 1000% and the tensile strength was about 15 MPa to 30 MPa. Differences in the 3D printed PBAT samples compared with previous studies can be attributed to several factors. Firstly, it is notable that any 3D printing techniques produce parts with anisotropic properties, meaning that the mechanical properties can vary based on the orientation in which the part is printed [38,39]. The specific printing parameters used in this study, such as the variation in the nozzle temperature, may have influenced the material’s molecular structure and chain alignment during the printing process. This could have resulted in enhanced elongation properties. An increase in temperature causes a decrease in the strength of the melt and a decrease in the viscosity of the mortar. According to previous sources [30], with the decrease in viscosity, the outflow from the nozzle increases, and this factor has an effect on the quality of printing. In addition, with the increase in temperature, the strength of the layers increases and a higher printing speed can be used. Also, the two parameters of speed and temperature strongly affect the stretching of polymer chains during extrusion. Reducing the temperature, increasing the speed, a low thickness of the layers, and a high cooling rate will stabilize the elongated shape of the chains in the direction of the movement of the nozzle and extrusion, and this phenomenon causes the stability and dimensional accuracy of the 3D printed parts. Of course, increasing the temperature greatly causes a drastic decrease in viscosity and makes it difficult to control the output flow. Therefore, choosing the optimal temperature is one of the most important challenges of printing new material.

On the other hand, 3D printing builds objects by depositing material layer by layer. The bonding between these layers may be weaker compared with the solid structure of traditionally manufactured parts. This weaker layer adhesion can result in a reduction in the overall tensile strength [40]. The Young’s modulus is a measure of a material’s stiffness, indicating its resistance to deformation under applied stress [41]. In this context, it signifies the ability of a printed part to maintain its shape and structural integrity when subjected to external forces. The consistent slope of the elastic region across all samples suggests that the Young’s modulus remains relatively unaffected by variations in nozzle temperature. This finding implies that changes in nozzle temperature between these three temperatures during the 3D printing process have a minimal impact on the stiffness or rigidity of the printed parts.

### 3.3. Microstructure

The scanning electron microscopy (SEM) analysis, depicted in Figure 7 and Figure 8, provides valuable insights into the printability and printing quality of the PBAT material, as well as the impact of the nozzle temperature on the printing process.

Figure 7 displays a cross-section of freeze-fractured printed samples, highlighting the adhesion between the printed layers and beads. This image enables us to investigate the quality of the printing process, specifically focusing on the effects of nozzle temperature on printing outcomes. It is worth emphasizing that the mechanical properties of FDM parts, such as elongation at break and tensile strength, are significantly influenced by both material and process parameters.

Among the process parameters, nozzle temperature and print speed play crucial roles, as they greatly influence the temperature profile during the printing process, encompassing the heating and cooling cycles [42,43]. Consequently, these factors impact the bonding strength between the deposited layers. With the circular cross-section of beads, the presence of undesired cavities between layers is inevitable, making these cavities potential sites for crack initiation and failure, particularly under tension. Raster breaking is the predominant failure mode in such cases. However, as the printability and printing quality improve, the size of these cavities tends to decrease, serving as an indicator of improved printability and printing quality [44].

Examining Figure 7, it is evident that all samples exhibited desirable cross-sections, with the printed beads and layers exhibiting strong bonding. This characteristic can be considered one of the reasons behind the desirable mechanical properties of the printed parts. The successful printing process by the modified FDM printer and the material’s good flowability were influential factors leading to this outcome. In Figure 7a, the size of the cavities between rasters is larger compared with that in Figure 7b,c, meaning that the integrity of the printed sample was lower, leading to weaker mechanical behavior. The largest cavity sizes observed in our printed samples were 840 µm, 430 µm, and less than 150 µm for printing temperatures of 160 °C, 180 °C, and 200 °C, respectively. For a more comprehensive comparison of cavity sizes with a closer view, please refer to Figure 8. It becomes apparent that, as the nozzle temperature increased, the size of the cavities decreased, thereby enhancing the integrity of the printed sample. This, in turn, contributed to improved mechanical properties, as mentioned earlier. The SEM images in Figure 7c indicate that a nozzle temperature of 200 °C produced the most uniform samples. Increasing the nozzle temperature in the printing process led to an improvement in mechanical properties for several reasons. As the nozzle temperature rose, the deposited material had a higher temperature when it came into contact with the previously printed layer. This elevated temperature promoted better fusion and bonding between the layers, resulting in stronger interlayer adhesion. Consequently, the printed part exhibited improved mechanical strength and resistance to delamination or layer separation. An increase in nozzle temperature can help to mitigate the formation of undesired cavities or voids between printed layers. These cavities can act as stress concentrators and potential initiation sites for cracks or failures [45]. By raising the nozzle temperature, the material becomes more fluid and can better fill in the gaps, reducing the size and extent of these cavities. As a result, the integrity and structural soundness of the printed part improve, leading to enhanced mechanical properties. Zhou et al. [46] employed an IR sensor to capture the temperature history. Their study demonstrated that higher nozzle and platform temperatures had the effect of extending diffusion time. As a consequence, this led to enhanced bond strength and overall mechanical properties.

It is important to note that, based on previous studies, the relationship between nozzle temperature and mechanical properties may have an optimal range. An excessive temperature increase beyond this range can lead to other issues, such as material degradation, excessive shrinkage or warping, or decreased dimensional accuracy. The adverse effects of high extrusion temperatures were studied by Ning et al. [47]. In their study, carbon fiber-reinforced polymer (with ABS as the matrix material) was utilized to investigate the impact of process parameters on the standard specimen. SEM analysis was employed to examine the fracture surfaces. The findings revealed that an increase in extrusion temperature resulted in improved part strength. However, once the temperature exceeded 220 °C, the mechanical properties experienced a sudden decline.

The SEM images presented in Figure 8 offer a magnified perspective on the cavity sizes within the printed PBAT material. These cavity sizes are clearly discernible, enabling a more detailed examination of their dimensions. Additionally, the SEM images effectively illustrate the consistent and uniform morphology of the printed PBAT, highlighting the homogeneity of its structure.

### 3.4. Energy Absorption and Compressive Behavior

The stress–strain curves of the printed cubic energy absorption samples, varying in infill percentage, are presented in Figure 9. These samples were subjected to compression testing, allowing for an examination of how the printed PBAT material responds to compressive loading and how the infill percentage influences its energy absorption and mechanical properties. By analyzing the stress–strain curves, valuable insights regarding the material’s mechanical behavior under compression can be gained. Figure 9 demonstrates that the PBAT printed structures exhibited favorable compressive properties and possessed an effective energy absorption ability. Additionally, it is evident that modifying the infill density during the printing process significantly enhanced the material’s mechanical properties in compression. Specifically, the initial peak of the stress–strain curve occurred at higher stresses for samples printed with a 60% infill density (1338 KPa), while the stress value was lower for samples printed with a 40% infill density (1306 KPa). This indicates that the structure of the 60% infill density sample was capable of withstanding higher stresses before yielding owing to the larger volume of material present within the cubic structure.

Furthermore, it is noteworthy that the 60% infill sample exhibited yielding at 50% strain, whereas the 40% infill sample yielded at 54% strain. Consequently, the 60% infill sample yielded slightly earlier than the latter. This difference may be attributed to a higher probability of defects existing in samples with higher infill density values, which could have influenced the overall yielding behavior of the material. Increasing the constituent mass directly affected the mechanical properties and its strengthening effect was evident. Therefore, increasing the infill density from 40% to 60%, taking into account all other conditions (the same printing parameters, geometry, and test parameters), increased the yield strength.

The results of the compression tests revealed a distinct correlation between the infill density of the samples and the slope of the stress–strain curves prior to the occurrence of the first peak. Notably, samples with higher infill densities exhibited steeper slopes, indicative of greater stiffness, whereas the samples with lower infill densities displayed shallower slopes, indicating lower stiffness. This implies that the infill density serves as a key determinant in shaping the material’s mechanical properties, specifically its stiffness. Consequently, a higher infill density can significantly enhance the material’s capacity to withstand compressive forces before yielding.

A key observation from the compression test results is that the overall shape of the stress–strain diagram remained the same for the material, regardless of how densely it was filled [48]. As a grid pattern was used, there was only one sharp peak in the diagram. This suggests that the overall shape of the stress–strain curve was independent of the infill density.

The inclusion of a higher infill density can also contribute to the material’s enhanced ability to absorb energy when subjected to compression loadings. Compression causes deformation in structures, resulting in energy absorption. The amount of energy absorbed by the structure can be determined by analyzing the stress–strain curve, which illustrates the relationship between the applied stress and resulting strain. The area under the stress–strain curve represents the energy absorbed by the structure during compression. Figure 9 indicates that PBAT with a higher infill density exhibited improved energy absorption during compression. This was because the denser structure allowed for higher stress levels, leading to a larger area under the stress–strain curve until a specific strain was reached. This finding is significant in understanding the mechanical properties of printed structures and can aid in the design of structures capable of absorbing energy under compression conditions.

A noteworthy finding is that, beyond the yield point, the stress–strain curve associated with the sample with 60% infill consistently remained positioned above the curve corresponding to the 40% infill sample. Furthermore, the area under the stress–strain curve for the 60% infill sample continued to be larger, even after yielding occurred, indicating that it had a greater capacity for absorbing energy. Also, the amounts of absorbed energy (area under the force–displacement curve) up to 100% strain for the 3D printed PBAT parts with 40% and 60% infill densities were 3.58 kJ and 4.08 kJ, respectively. Qualitatively, it is clear that, with the increase in infill density, the amount of area under the curve increased.

### 3.5. Ability to Maintain Shape

The average expansions in the three dimensions of length, width, and height for scaffolds printed with 40% infill density were calculated as 0.65 mm, 0.49 mm, and 0.41 mm, respectively. These values decreased by increasing the amount of infill from 40% to 60% and reached 0.41 mm, 0.33 mm, and 0.28 mm, respectively. These values are quite promising compared with those of conventional materials, such as PLA, with 100% infill and indicate the high dimensional accuracy of porous PBAT printed parts with two different infill densities.

### 3.6. Advantages and Future of Direct Pellet-Based Printing

One of the main limitations in the new field of 3D printing, particularly when using biodegradable materials, is the restriction in filament production and printing capabilities. As a result, the selection of 3D printing materials is limited to a few commercially available thermoplastics that have suitable printing capabilities, such as PLA, ABS, PETG, PCL, PC, and PA. The use of commercial FDM mechanisms is restricted to these polymers. However, with the method used, almost all materials can be printed in the granule form, eliminating need for filament production. Another issue is the high cost of filament production, which includes expenses for the additive to enhance the extrudability and printability of thermoplastics. For instance, the TPU filaments available on the market contain more than 1.5% carbon black. When considering all aspects of the direct granule printing method compared with commercial FDM, it becomes evident that it is not limited to specific materials and its raw material in the form of granules, which is much more affordable than filament. Additionally, this method allows for the printing of various thermoplastic elastomers, composites, and blends. Consequently, it is expected that the use of this method for printing more practical materials will expand, allowing for a wide range of diverse mechanical, physical, and biological properties to be obtained with the help of the 3D printing process in the future. This opens up opportunities for various industries, including automotive, aerospace, and medical, to explore new possibilities in design and manufacturing.

## 4. Conclusions

PBAT’s biodegradability and compostability make it a suitable material for environmentally friendly 3D printing. By incorporating PBAT into the 3D printing process, it opens up possibilities for the production of various objects with a reduced ecological footprint. This study successfully demonstrated the feasibility of direct pellet printing of PBAT without the need for filament conversion. By customizing the printer with a pneumatically operated pellet hopper and feeding system, accurate and controlled feeding of polymer pellets was achieved. This approach has the potential to simplify the PBAT 3D printing process and reduce material waste compared with conventional filament-extrusion methods. In this study, we printed three sets of tensile specimens using different nozzle temperatures to identify the optimal temperature for printing PBAT. The mechanical properties and microstructure of the printed specimens were investigated. Additionally, dynamic mechanical thermal analysis was employed to examine the thermal behavior of the printed PBAT. Furthermore, we designed and printed two energy-absorption structures with optimized parameters, including different infill percentages. The compressive and energy-absorption behaviors of the material were studied using these structures. Several conclusions can be drawn regarding the use of PBAT as a biodegradable material for 3D printing:The dynamic mechanical thermal analysis (DMTA) revealed that PBAT undergoes a transition from a glassy state to a rubbery state at a specific temperature (about −25 °C). This characteristic, along with its storage modulus and tan δ values, indicates that PBAT possesses elastomeric and soft properties at room temperature, making it suitable for flexible and resilient applications and hard to 3D print with conventional filament-based processes;The mechanical properties of the printed PBAT specimens were influenced by the nozzle temperature. Higher nozzle temperatures resulted in improved mechanical properties, including increased elongation at break and tensile strength. The sample printed at the highest nozzle temperature of 200 °C exhibited the best performance, with a remarkable elongation of 1379% and a tensile strength of 7.5 MPa;The 3D printed PBAT structures showed promising energy-absorption behavior. The specimens with 60% infill density demonstrated higher compressive strength (1338 KPa) and energy absorption compared with those with 40% infill density (1306 KPa). This suggests that PBAT has the potential to be utilized in applications where impact resistance and energy absorption are crucial;The SEM images showed that, with the increase in printing temperature, the quality of the PBAT printed parts improved significantly, in a way; for the sample printed at 160 °C, the microholes and weak adhesion between the layers are quite clear. meanwhile, the volume of microholes decreased with the increase in temperature up to 180 °C, and for the sample printed at 200 °C, the highest print quality was achieved and it was difficult to find microholes or even a layered structure;Overall, this study demonstrates the potential of PBAT as a standalone material for 3D printing applications, showcasing its unique properties, printing feasibility, and desirable mechanical performance. Further research and development in this area could contribute to the advancement of sustainable and eco-friendly 3D printing practices.

## Figures and Tables

**Figure 1 polymers-16-00267-f001:**
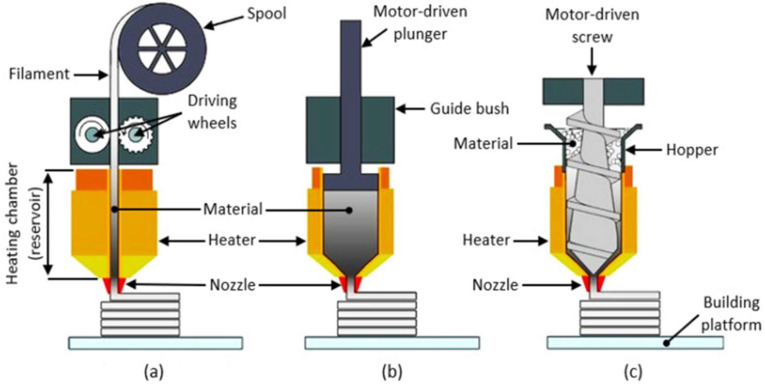
Material extrusion processes: (**a**) filament-based, (**b**) plunger or syringe-based, and (**c**) screw-based extrusion [29].

**Figure 2 polymers-16-00267-f002:**
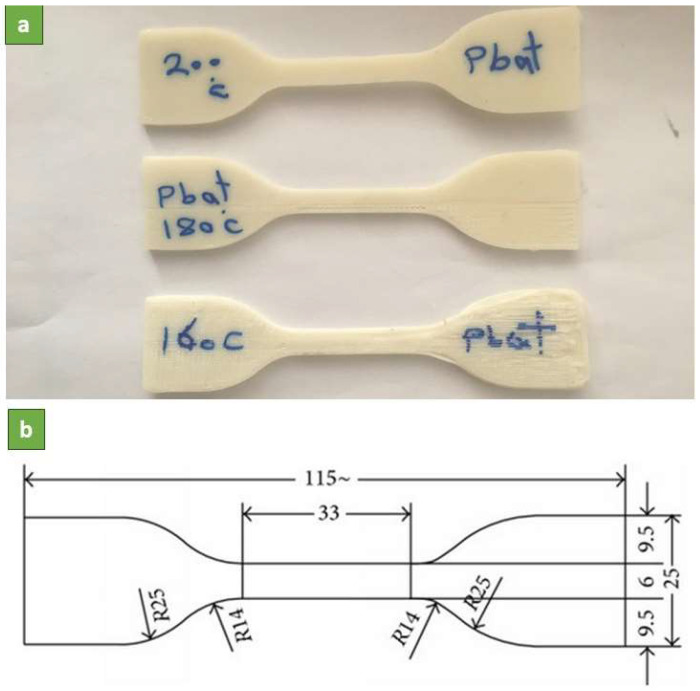
Printed tensile samples: (**a**) real samples and (**b**) standard dimensions (unit: mm).

**Figure 3 polymers-16-00267-f003:**
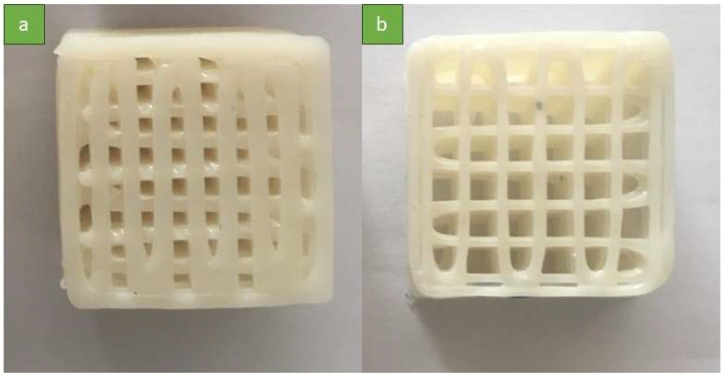
Printed energy-absorption samples: (**a**) 60% infill density and (**b**) 40% infill density.

**Figure 4 polymers-16-00267-f004:**
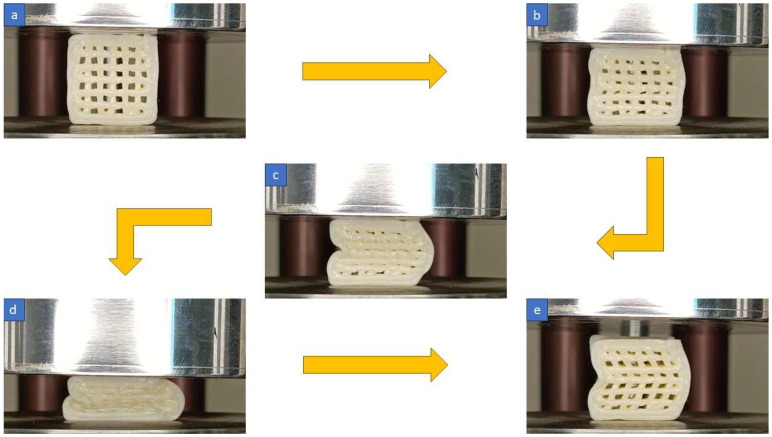
Energy absorption process (**a**) before the test, (**b**,**c**) during the test, (**d**) at the end of the test, and (**e**) after the test.

**Figure 5 polymers-16-00267-f005:**
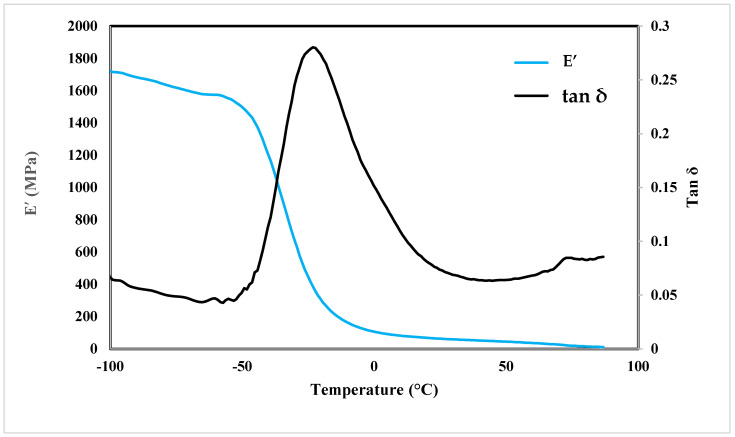
Dynamic mechanical thermal analysis results.

**Figure 6 polymers-16-00267-f006:**
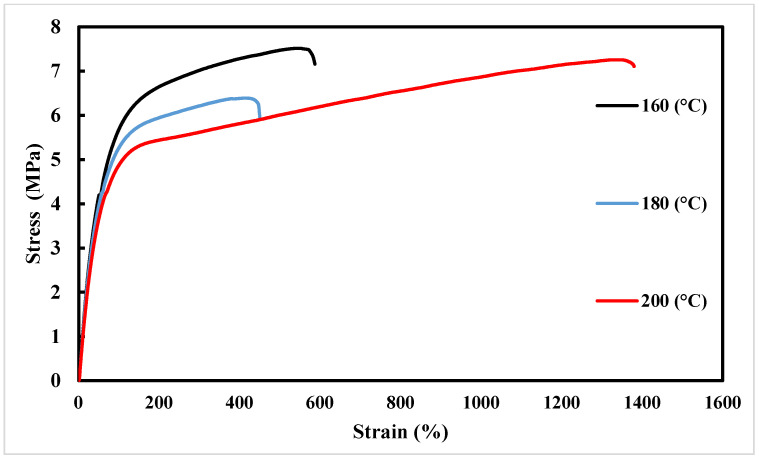
Tensile test results for different nozzle temperatures.

**Figure 7 polymers-16-00267-f007:**
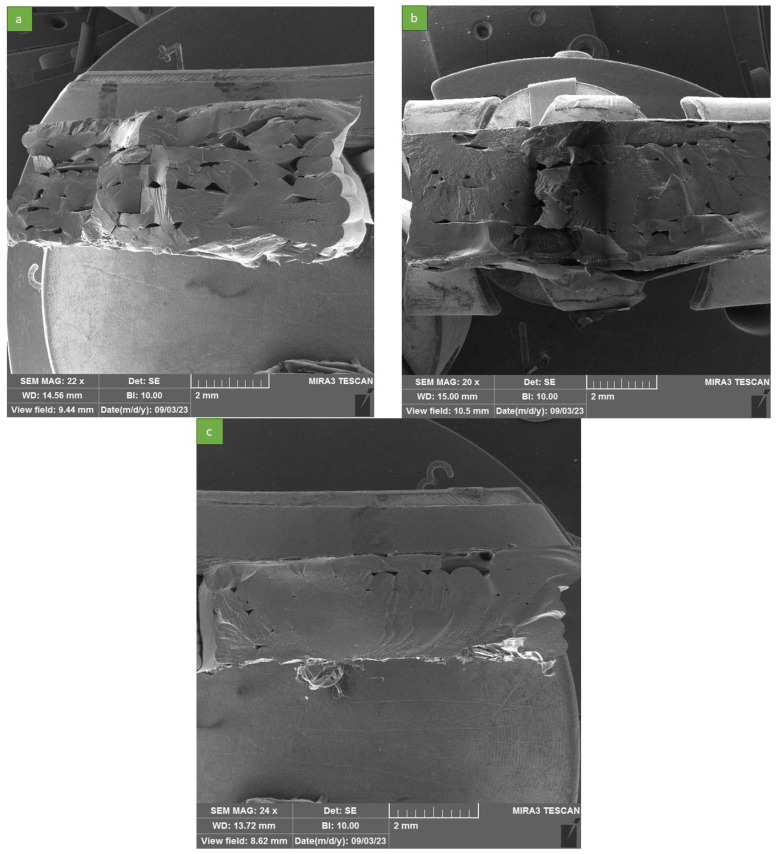
SEM images from interlayer adhesion for different nozzle temperatures: (**a**) 160 °C, (**b**) 180 °C, and (**c**) 200 °C.

**Figure 8 polymers-16-00267-f008:**
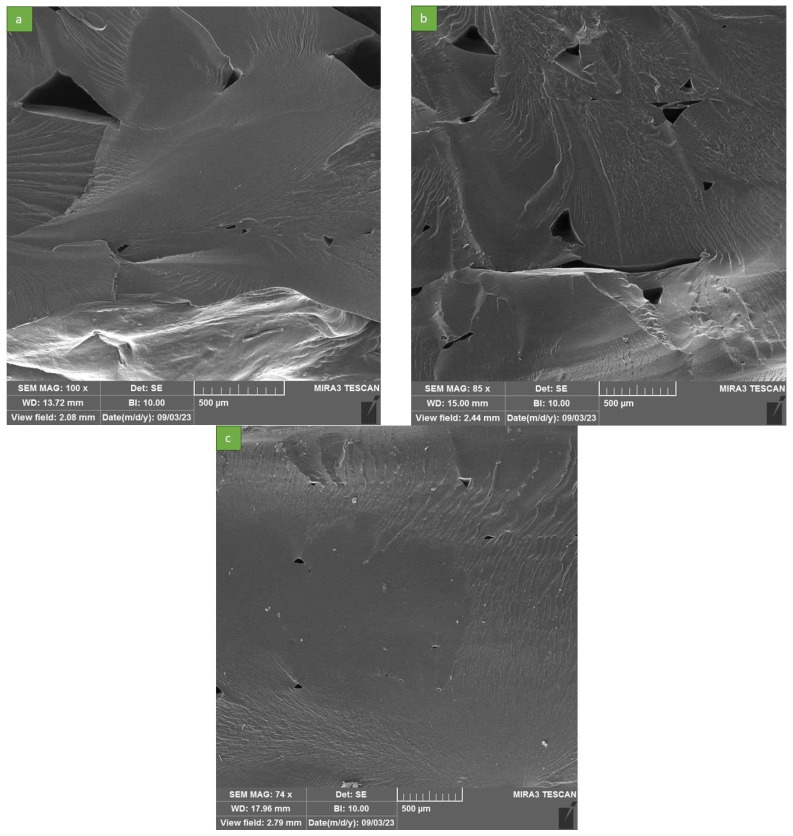
SEM images from cavities and PBAT morphology for different nozzle temperature: (**a**) 160 °C (**b**) 180 °C and (**c**) 200 °C.

**Figure 9 polymers-16-00267-f009:**
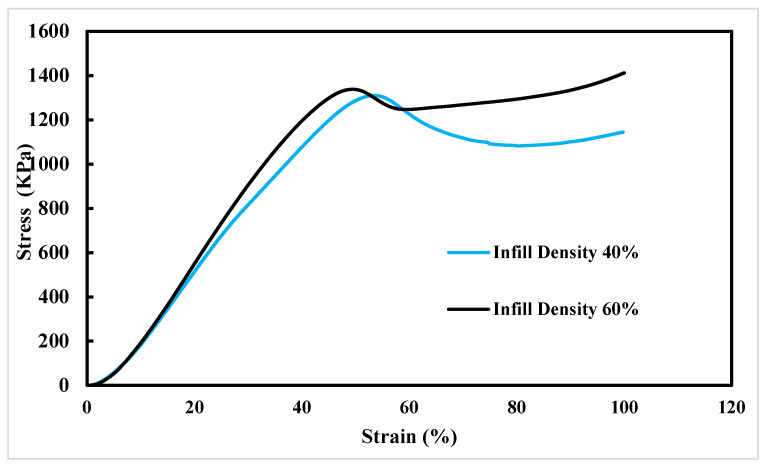
Stress–strain curves for 3D printed PBAT parts with 40% and 60% infill densities under compression loading.

**Table 1 polymers-16-00267-t001:** Printing parameters.

Nozzle temperature (°C)	160–180–200
Nozzle diameter (mm)	0.6
Printing speed (mm/min)	300
Layer height (mm)	0.4
Raster angles (°)	0/90
Infill density (%)	40, 60, and 100
Shell number	2

## Data Availability

Data are contained within the article.

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
