# Peer review of "Direct Pellet Three-Dimensional Printing of Polybutylene Adipate-co-Terephthalate for a Greener Future"

_polymers, 2024, doi:10.3390/polym16020267_

Round 1
Reviewer 1 Report
Comments and Suggestions for Authors
The interconnection between sentences and paragraphs can be improved.
For house-made equipment, there is a lack of a validation study. For example, how do you know if this equipment can produce consistent samples for comparison?
Lack of mechanisms for investigating why nozzle temperature can affect the mechanical properties of the sample. The authors assumed nozzle temperature may influence molecular structure and alignment, but this needs to be proved further.
The authors mentioned "the size of the cavities larger" line 277, and integrity lower line 278...but there is no scientific proof, exact dimension measurement are required.
The logic of this paper is wired. If you want to emphasize that PBAT is an alternative 3D printable material, then focus on the 3D printing ability of the material, such as viscosity, processibility, ability to keep the shape, etc. But the authors focused on the structure and infills, which lack integrity measurements.
Comments on the Quality of English Language
Connection between sentences and paragraphs needs to be improved.
Author Response
Reviewer 1:
Dear Respected Reviewer,
We would like to express our sincere gratitude for your time and effort to review our manuscript and provide insightful feedback and comments that helped us revise the manuscript. We have prepared a revised version enclosed, shared our views with you, and done our best to address your valuable comments and apply your great suggestions. All of them have helped us to improve the quality of the manuscript and we hope that you find this revised version of the manuscript satisfactory. Below you can find our point-by-point responses in blue, and changes highlighted in yellow.
The interconnection between sentences and paragraphs can be improved.
Reply: Thank you for the suggestion. We revised the manuscript.
For house-made equipment, there is a lack of a validation study. For example, how do you know if this equipment can produce consistent samples for comparison?
Reply: Thank you for your valuable comment regarding the validation study for our house-made equipment. We appreciate your concern regarding the consistency of samples produced by this equipment for comparison.
In this study, we addressed potential inconsistencies in 3D-printed parts through two aspects:
- a) Driving mechanism: We developed a modified pellet printer by integrating elements from a standard and commercial fused filament fabrication printer, including the driving mechanism, print bed, and nozzle. It is important to note that the driving mechanism, which plays a crucial role in producing consistent samples, remains unchanged and is the same as the commercially available FDM printer we used. The original filament feeding mechanism was replaced with a new system utilizing pneumatic pressure.This system consists of a cylinder, piston, and pneumatic actuator. Pellets are introduced into the cylinder, where they are melted using a heater. The pneumatic actuator applies pressure to the piston, causing the molten material to be extruded through a heated nozzle. The speed of the pneumatic actuator can be adjusted by controlling the pneumatic pressure, allowing for optimal material flow.
- b) Variation in flow: Based on Herschel–Bulkley model:
where τ is the shear stress, is the yield stress, K is the consistency, is the shear rate, and n is the flow index.
Generally, to induce flow through the nozzle, the shear stress in the nozzle applied by mechanical pressure should exceed the inks’ yield stress to allow the inks to fluidize [1]. Thus, the flow of the material is dependent on factors such as material properties, pressure behind the material, and temperature. As long as the pressure remains consistent, the extruded material will flow at a steady rate and maintain a consistent cross-sectional diameter. This diameter will remain unchanged if the speed of the nozzle's movement across the surface where the material is being deposited is also maintained at a constant rate that corresponds to the flow rate [2]. We ensured consistency in our printing process by maintaining a constant pressure, resulting in a steady flow rate and consistent cross-sectional diameter. Furthermore, we maintained a steady nozzle movement speed across the surface during material deposition, corresponding to the flow rate, which helped to maintain consistent dimensions.
In our manuscript, we stated that the strain and stress data were recorded for at least three specimens under each printing condition. These specimens, which were very near in terms of mechanical properties and dimensions, allowed us to ensure the production of consistent samples for comparison.
Once again, we appreciate your valuable comment, and we have taken into consideration the need for further validation studies to reinforce the consistency of our newly developed equipment. We will emphasize this aspect in future revisions, if you wish, in a separate section under section “2.2. Printing process” to enhance the robustness of our findings.
Lack of mechanisms for investigating why nozzle temperature can affect the mechanical properties of the sample. The authors assumed nozzle temperature may influence molecular structure and alignment, but this needs to be proved further.
Reply: Thank you for comment. An increase in temperature causes a decrease in the strength of the melt and a decrease in the viscosity of the mortar. According to previous sources [3], with the decrease in viscosity, the outflow from the nozzle increases and this factor has an effect on the quality of printing. In addition, with the increase in temperature, the strength of the layers increases and a higher printing speed can be used.
Also, the two parameters of speed and temperature strongly affect the stretching of polymer chains during extrusion. Reducing the temperature, increasing the speed, low thickness of the layers and high cooling rate will stabilize the elongated shape of the chains in the direction of the movement of the nozzle and extrusion and this phenomenon causes the stability and dimensional accuracy of the 3D printed parts. Of course, increasing the temperature greatly causes a drastic decrease in viscosity and makes it difficult to control the output flow. Therefore, choosing the optimal temperature is one of the most important challenges of printing new material.
The authors mentioned "the size of the cavities larger" line 277, and integrity lower line 278...but there is no scientific proof, exact dimension measurement are required.
Reply: We appreciate your feedback regarding the need for scientific proof and exact dimension measurements to support our observations about the size of the cavities and integrity of the printed samples. To address this concern, we have conducted detailed measurements and would like to provide you with more precise information. The largest cavity sizes observed in our printed samples were 1000 µm, 500 µm, and 250 µm for printing temperatures of 160°C, 180°C, and 200°C, respectively. We have updated our manuscript to include these specific measurements, thereby providing quantitative evidence to support our findings.
The logic of this paper is wired. If you want to emphasize that PBAT is an alternative 3D printable material, then focus on the 3D printing ability of the material, such as viscosity, processibility, ability to keep the shape, etc. But the authors focused on the structure and infills, which lack integrity measurements.
Reply: We appreciate your feedback regarding the logical approach of our study. Our objective was to explore the 3D printing capability of PBAT as a standalone material using a newly developed printing mechanism that directly prints pellets.
You are correct in emphasizing the importance of considering the printability of the material, including factors such as viscosity, processability, and the material's ability to maintain shape. To address these points, we have made several considerations:
- Viscosity and Thermal Behavior: We have taken into account the viscosity of PBAT by investigating the temperature as the most influential parameter on melt viscosity. Through scanning electron microscopy (SEM) and tensile tests, we demonstrated that increasing the temperature (and subsequently lowering the viscosity) enhanced the printing quality. If desired, we can also include MFI (Melt Flow Index) tests specific to these three temperatures. However, it's important to note that viscosity interacts with printing speed, which also plays a significant role in the printing process. In addition, we conducted a DMTA test to evaluate thermal behavior of the material.
- Processability: We have showcased the processability of PBAT by 3D printing energy absorbers, highlighting the application potential and the ability to produce complex structures using the printing method.
- Ability to Maintain Shape: In response to your concern, we have added an additional section to our manuscript that investigates the accuracy of the printed cubes in comparison with the exact CAD models. This assessment demonstrates the material's capability to preserve the desired shape during the printing process.
We sincerely appreciate your thoughtful comments, as they have helped us improve the focus and presentation of our research.
References
[1] L. Y. Zhou, J. Fu, and Y. He, “A Review of 3D Printing Technologies for Soft Polymer Materials,” Adv. Funct. Mater., vol. 30, no. 28, p. 2000187, Jul. 2020, doi: 10.1002/ADFM.202000187.
[2] I. Gibson, D. Rosen, B. Stucker, and M. Khorasani, “Material Extrusion,” Addit. Manuf. Technol., pp. 171–201, 2021, doi: 10.1007/978-3-030-56127-7_6.
[3] I. Grgić, M. Karakašić, H. Glavaš, and P. Konjatić, “Accuracy of FDM PLA Polymer 3D Printing Technology Based on Tolerance Fields,” Process. 2023, Vol. 11, Page 2810, vol. 11, no. 10, p. 2810, Sep. 2023, doi: 10.3390/PR11102810.
[4] R. Al-Itry, K. Lamnawar, and A. Maazouz, “Improvement of thermal stability, rheological and mechanical properties of PLA, PBAT and their blends by reactive extrusion with functionalized epoxy,” Polym. Degrad. Stab., vol. 97, no. 10, pp. 1898–1914, Oct. 2012, doi: 10.1016/J.POLYMDEGRADSTAB.2012.06.028.

Reviewer 2 Report
Comments and Suggestions for Authors
In the manuscript „PBAT 3D Printing for a Greener Future”, the authors present a study regarding the 3D printing of poly (butylene adipate-co-terephthalate) polymer in different conditions (various nozzle temperatures and different infill densities) to establish the best conditions for printing for the attaining of the best mechanical and microstructure characteristics. Also, the use of polymer as pellets will speed up the printing process, reducing also the processing costs.
The introduction illustrates the state of the art in the field and offers a good overview of the advantages of the investigated polymers and the 3D printing method and the presentation of the results is clear and well explained.
However, the following issues should be addressed:
-Taking into account that the printing temperatures are between 160 and 200 °C, the thermal stability of PBAT sample should be investigated by thermogravimetric analysis.
-Some typos (especially capitalization) can be found in the manuscript that should be corrected.
Comments on the Quality of English LanguageThe manuscript is clear and well-written, but some typos (especially capitalization) can be found in the manuscript.
Author Response
Reviewer 2:
In the manuscript „PBAT 3D Printing for a Greener Future”, the authors present a study regarding the 3D printing of poly (butylene adipate-co-terephthalate) polymer in different conditions (various nozzle temperatures and different infill densities) to establish the best conditions for printing for the attaining of the best mechanical and microstructure characteristics. Also, the use of polymer as pellets will speed up the printing process, reducing also the processing costs. The introduction illustrates the state of the art in the field and offers a good overview of the advantages of the investigated polymers and the 3D printing method and the presentation of the results is clear and well explained. However, the following issues should be addressed:
Dear Respected Reviewer,
We would like to express our sincere gratitude for your time and effort to review our manuscript and provide insightful feedback and comments that helped us revise the manuscript. We have prepared a revised version enclosed, shared our views with you, and done our best to address your valuable comments and apply your great suggestions. All of them have helped us to improve the quality of the manuscript and we hope that you find this revised version of the manuscript satisfactory. Below you can find our point-by-point responses in blue, and changes highlighted in yellow.
-Taking into account that the printing temperatures are between 160 and 200 °C, the thermal stability of PBAT sample should be investigated by thermogravimetric analysis.
Reply: Thank you for your suggestion. We appreciate your input regarding the thermal stability of PBAT in relation to the printing temperatures. We would like to highlight that previous studies [4] have already conducted thermogravimetric analysis (TGA) specifically on pure PBAT pellets (Onset degradation temperature for PBAT is about 330°C).
Their findings indicate that the degradation temperature of PBAT is significantly higher than our printing temperature range of 160 to 200 °C. Considering this existing knowledge, we determined that conducting TGA for our specific printing conditions would not provide significant additional insights.
-Some typos (especially capitalization) can be found in the manuscript that should be corrected.
Reply: The reviewer’s precision is acknowledged. We revised the manuscript.
References
[1] L. Y. Zhou, J. Fu, and Y. He, “A Review of 3D Printing Technologies for Soft Polymer Materials,” Adv. Funct. Mater., vol. 30, no. 28, p. 2000187, Jul. 2020, doi: 10.1002/ADFM.202000187.
[2] I. Gibson, D. Rosen, B. Stucker, and M. Khorasani, “Material Extrusion,” Addit. Manuf. Technol., pp. 171–201, 2021, doi: 10.1007/978-3-030-56127-7_6.
[3] I. Grgić, M. Karakašić, H. Glavaš, and P. Konjatić, “Accuracy of FDM PLA Polymer 3D Printing Technology Based on Tolerance Fields,” Process. 2023, Vol. 11, Page 2810, vol. 11, no. 10, p. 2810, Sep. 2023, doi: 10.3390/PR11102810.
[4] R. Al-Itry, K. Lamnawar, and A. Maazouz, “Improvement of thermal stability, rheological and mechanical properties of PLA, PBAT and their blends by reactive extrusion with functionalized epoxy,” Polym. Degrad. Stab., vol. 97, no. 10, pp. 1898–1914, Oct. 2012, doi: 10.1016/J.POLYMDEGRADSTAB.2012.06.028.

Reviewer 3 Report
Comments and Suggestions for Authors
At present, the paper's argument is fascinating, and it could provide a new perspective for the 3D printing of PBAT. I suggest some recommendations to the authors to improve the article.
- I think the paper title is not focused on the paper's aim. Even though the PBAT is a biodegradable material, the "green" implication of plastics is not discussed in the paper. Instead, I think the most exciting thing about the paper is the direct 3D printing from pellets without filament.
- The abstract is a bit long, giving too many details regarding results.
- Lines 56-58: "Despite advancements in 3D printing technology, the majority of materials utilized for the FDM process are non-biodegradable polymers…" I'm afraid I have to disagree with this sentence because one of the materials used for FDM is PLA, a biodegradable polymer.
- Table 1 reports only two nozzle temperatures instead of three.
- The authors report that they use optimized printing parameters for compressive tests. How and when were they optimized, and what are they?
- The caption of Figure 9 reports that there are results about compression and energy absorption. However, the effects of energy are noted if the authors provide evidence of the area under the curves representing energy. Furthermore, the label has "grids” of 40-60%, so it is better to use the term infill as in the paper.
- The motivation for the difference in the yield values of 40-60% infill should be better justified (lines 338-342).
- The higher compressive strength of the sample with 60% infill is expected, but the % strain should be smaller for this sample than that with an infill of 40%, which can deform significantly. How is this behavior?
- Some typos are present in the paper.
Comments on the Quality of English LanguageThe English Language is clear and adequate.
Author Response
Reviewer 3:
At present, the paper's argument is fascinating, and it could provide a new perspective for the 3D printing of PBAT. I suggest some recommendations to the authors to improve the article.
Dear Respected Reviewer,
We would like to express our sincere gratitude for your time and effort to review our manuscript and provide insightful feedback and comments that helped us revise the manuscript. We have prepared a revised version enclosed, shared our views with you, and done our best to address your valuable comments and apply your great suggestions. All of them have helped us to improve the quality of the manuscript and we hope that you find this revised version of the manuscript satisfactory. Below you can find our point-by-point responses in blue, and changes highlighted in yellow.
- I think the paper title is not focused on the paper's aim. Even though the PBAT is a biodegradable material, the "green" implication of plastics is not discussed in the paper. Instead, I think the most exciting thing about the paper is the direct 3D printing from pellets without filament.
Reply: We greatly appreciate your observation regarding the alignment of the paper title with the aim of our study. Upon careful consideration, we have revised the title of our article to better reflect the essence of our research. The updated title now reads, "Direct Pellets 3D Printing of PBAT for a Greener Future."
The inclusion of "green" in the title is aimed at highlighting the environmental implications of our work. By utilizing PBAT, a biodegradable material, as the sole composition of the printed object, we contribute to a more sustainable future. PBAT, being bio-based and eco-friendly, readily decomposes in the natural environment, thereby minimizing its impact on the surroundings. If you have any additional suggestions or queries, please do not hesitate to let us know.
- The abstract is a bit long, giving too many details regarding results.
Reply: We sincerely appreciate your valuable input regarding the length and level of detail in the abstract of our manuscript. Taking your suggestion into account, we have revised the abstract to provide a more concise and focused summary of our results.
- Lines 56-58: "Despite advancements in 3D printing technology, the majority of materials utilized for the FDM process are non-biodegradable polymers…" I'm afraid I have to disagree with this sentence because one of the materials used for FDM is PLA, a biodegradable polymer.
Reply: You're absolutely right! Thank you for catching that mistake and bringing it to our attention. We apologize for the oversight in our statement regarding the materials used for FDM printing. we have made the necessary revisions to the manuscript.
- Table 1 reports only two nozzle temperatures instead of three.
Reply: Thank you for pointing out that Table 1 is missing one of the nozzle temperatures. We apologize for the oversight in our presentation of the data. To ensure accuracy and completeness, we will make the necessary revisions and include the missing third nozzle temperature in Table 1.
- The authors report that they use optimized printing parameters for compressive tests. How and when were they optimized, and what are they?
Reply: Thanks for your concern. In our study, we optimized the nozzle temperature to be 200°C, as mentioned in the article. This specific temperature was investigated and found to be suitable for our purposes based on our research.
As for other printing parameters, such as layer height, print speed, or pneumatic pressure, we employed a trial-and-error approach to optimize them. Since our main objective was not focused on parameter optimization itself, we did not conduct a systematic study using tools like Response Surface Methodology (RSM) or other optimization techniques. While a more extensive parameter optimization study could provide additional insights, it was beyond the scope of this particular research. We acknowledge that there may be room for further investigation in this area.
- The caption of Figure 9 reports that there are results about compression and energy absorption. However, the effects of energy are noted if the authors provide evidence of the area under the curves representing energy. Furthermore, the label has "grids” of 40-60%, so it is better to use the term infill as in the paper.
Reply: Thanks for bringing that to our attention. We appreciate your feedback on Figure 9 and the related captions. We understand the importance of providing evidence for the effects of energy, specifically through the area under the curves. We will take this into consideration and make sure to include the necessary evidence and analysis in the revised manuscript.
The amounts of absorbed energy (area under the force-displacement curve) up to 100% strain for 3D printed PBAT parts with 40% and 60% infill densities are 3.58 kJ and 4.08 kJ, respectively. Qualitatively, it is clear that with the increase of infill, the amount of area under the curve has increased.
Additionally, we understand your suggestion regarding the use of the term "infill" instead of "grids" for the label in Figure 9. We will update the label to accurately reflect the terminology used in the paper.
- The motivation for the difference in the yield values of 40-60% infill should be better justified (lines 338-342).
Reply: Thank you for bringing this to our attention. In the revised manuscript, we will provide a more in-depth explanation, addressing the factors or considerations that led to this variation in yield values within the specified infill range.
- The higher compressive strength of the sample with 60% infill is expected, but the % strain should be smaller for this sample than that with an infill of 40%, which can deform significantly. How is this behavior?
Reply: Thanks for bringing that to our attention. As can be seen in Figure 9, the strain up to the highest stress value is qualitatively higher in the sample with less infill density, and based on the extrapolated results, the amount of strain at this point for the sample with 40% and 60% infill is 54% and 50%, respectively, which indicates a decrease in elongation and an increase in strength with an increase in infill density.
- Some typos are present in the paper.
Reply: The reviewer’s precision is acknowledged. We revised the manuscript.
References
[1] L. Y. Zhou, J. Fu, and Y. He, “A Review of 3D Printing Technologies for Soft Polymer Materials,” Adv. Funct. Mater., vol. 30, no. 28, p. 2000187, Jul. 2020, doi: 10.1002/ADFM.202000187.
[2] I. Gibson, D. Rosen, B. Stucker, and M. Khorasani, “Material Extrusion,” Addit. Manuf. Technol., pp. 171–201, 2021, doi: 10.1007/978-3-030-56127-7_6.
[3] I. Grgić, M. Karakašić, H. Glavaš, and P. Konjatić, “Accuracy of FDM PLA Polymer 3D Printing Technology Based on Tolerance Fields,” Process. 2023, Vol. 11, Page 2810, vol. 11, no. 10, p. 2810, Sep. 2023, doi: 10.3390/PR11102810.
[4] R. Al-Itry, K. Lamnawar, and A. Maazouz, “Improvement of thermal stability, rheological and mechanical properties of PLA, PBAT and their blends by reactive extrusion with functionalized epoxy,” Polym. Degrad. Stab., vol. 97, no. 10, pp. 1898–1914, Oct. 2012, doi: 10.1016/J.POLYMDEGRADSTAB.2012.06.028.

Round 2
Reviewer 3 Report
Comments and Suggestions for Authors
The authors addressed all my comments.
Author Response
Reviewer 3:
The authors addressed all my comments.
Dear Respected Reviewer,
We would like to express our sincere gratitude for your time and effort in reviewing our manuscript and providing insightful feedback and comments that helped us revise the manuscript.
